Effect of erythromycin residuals in food on the development of resistance in Streptococcus pneumoniae: an in vivo study in Galleria mellonella

http://orcid.org/0009-0002-5790-0096 Baranchyk Yuliia 1 2
Gestels Zina 2
Van den Bossche Dorien 2
Abdellati Saïd 2
http://orcid.org/0000-0002-5897-0240 Britto Xavier Basil 2 3
http://orcid.org/0000-0001-8421-5137 Manoharan-Basil Sheeba Santhini 2
Kenyon Chris 2 4 ckenyon@itg.be
1 UnivLyon, Université Claude Bernard Lyon 1 , Lyon , France
2 Institute of Tropical Medicine Antwerp , Antwerp , Belgium
3 Hospital Outbreak Support Team-HOST, Ziekenhuis Netwerk Antwerpen Middelheim , Antwerp , Belgium
4 Department of Medicine, University of Cape Town , Cape Town , South Africa
Gillespie Joseph
Electronic publication date: 2024 May 28
Publication date: 2024
Volume: 12
Electronic Location ID: e17463
Received 2024 Jan 25; Accepted 2024 May 5
Copyright: © 2024 Baranchyk et al.
Copyright year: 2024
Copyright holder: Baranchyk et al.
License: This is an open access article distributed under the terms of the Creative Commons Attribution License, which permits unrestricted use, distribution, reproduction and adaptation in any medium and for any purpose provided that it is properly attributed. For attribution, the original author(s), title, publication source (PeerJ) and either DOI or URL of the article must be cited.
License URL: https://creativecommons.org/licenses/by/4.0/

Keywords: Streptococcus pneumoniae, Acceptable daily intake, AMR, Minimum selection concentration

Funding: SOFI 2021 grant—‘PReventing the Emergence of untreatable STIs via radical Prevention’ (PRESTIP) EMJMD LIVE (Erasmus+ Mundus Joint Master Degree Leading International Vaccinology Education) EACEA (Education, Audiovisual and Culture Executive Agency) of the European Commission 2018-1484 EACEA The study was funded by SOFI 2021 grant—‘PReventing the Emergence of untreatable STIs via radical Prevention’ (PRESTIP). Yuliia Baranchyk was registered in the EMJMD LIVE (Erasmus+ Mundus Joint Master Degree Leading International Vaccinology Education), co-funded by the EACEA (Education, Audiovisual and Culture Executive Agency, award 2018-1484) of the European Commission, and received a scholarship from the EACEA. The funders had no role in study design, data collection and analysis, decision to publish, or preparation of the manuscript.

==============================
Background

The use of antimicrobials to treat food animals may result in antimicrobial residues in foodstuffs of animal origin. The European Medicines Association (EMA) and World Health Organization (WHO) define safe antimicrobial concentrations in food based on acceptable daily intakes (ADIs). It is unknown if ADI doses of antimicrobials in food could influence the antimicrobial susceptibility of human-associated bacteria.

Objectives

This aim of this study was to evaluate if the consumption of ADI doses of erythromycin could select for erythromycin resistance in a Galleria mellonella model of Streptococcus pneumoniae infection.

Methods

A chronic model of S. pneumoniae infection in G. mellonella larvae was used for the experiment. Inoculation of larvae with S. pneumoniae was followed by injections of erythromycin ADI doses (0.0875 and 0.012 μg/ml according to EMA and WHO, respectively). Isolation of S. pneumoniae colonies was then performed on selective agar plates. Minimum inhibitory concentrations (MICs) of resistant colonies were measured, and whole genome sequencing (WGS) was performed followed by variant calling to determine the genetic modifications.

Results

Exposure to single doses of both EMA and WHO ADI doses of erythromycin resulted in the emergence of erythromycin resistance in S. pneumoniae. Emergent resistance to erythromycin was associated with a mutation in rplA, which codes for the L1 ribosomal protein and has been linked to macrolide resistance in previous studies.

Conclusion

In our in vivo model, even single doses of erythromycin that are classified as acceptable by the WHO and EMA induced significant increases in erythromycin MICs in S. pneumoniae. These results suggest the need to include the induction of antimicrobial resistance (AMR) as a significant criterion for determining ADIs.

Introduction

There is increasing evidence that low concentrations of antimicrobials can select for antimicrobial resistance (AMR). Studies have found that antimicrobial concentrations over 200-fold lower than the minimum inhibitory concentration (MIC) select for resistant vs susceptible strains of Escherichia coli and Salmonella enterica spp (Gullberg et al., 2014; Stanton et al., 2020). These studies have defined the minimal selective concentration (MSC) as the minimum concentration of an antimicrobial that selects for antimicrobial resistance (Gullberg et al., 2014, 2011). Two types of MSC have been defined. The MSCdenovo is defined as the minimum concentration of an antimicrobial at which one can induce de novo resistance. The MSCselect is the lowest antimicrobial concentration that selects for a resistant compared to a susceptible strain (Kraupner et al., 2020). Gullberg et al. (2011) found the E. coli ciprofloxacin MSCselect to be 230-fold lower than the MIC, and the MSCdenovo to be at least tenfold lower than the MIC.

Much remains unknown about the MSCs for macrolides. The E. coli erythromycin MSCselect has been found to be less than 0.200 µg/mL, which is less than 1/60th of the MIC (Gullberg et al., 2014). Stanton et al. (2020) determined the E. coli erythromycin MSCselect for the ermF gene in a complex microbial community to be a similar concentration, 0.514 µg/mL. They did however find that this concentration was 1–2 orders of magnitude greater than measured environmental macrolide concentrations (Stanton et al., 2020). Theoretical MSCs for the most susceptible species can be calculated by applying the ratio between the MIC and MSC from a species where this has been measured, such as E. coli, to the species with the lowest MIC for the antimicrobial in the EUCAST dataset (Bengtsson-Palme & Larsson, 2016; EFSA Panel on Biological Hazards (BIOHAZ) et al., 2021). This predicted MSC value (PMSC) has been calculated to be 0.13 µg/mL for erythromycin (EFSA Panel on Biological Hazards (BIOHAZ) et al., 2021). Macrolide MSCs have not been evaluated for other bacterial species and have never been assessed in vivo. Testing MSCs in vivo is particularly important since previous studies have found that MSCs are typically higher in complex polymicrobial environments (Stanton et al., 2020). A crucial hypothesis to test is if the concentrations of antimicrobials allowed in food can induce AMR in pathobionts colonizing humans. This hypothesis has been raised by authors who have found evidence that the consumption of antimicrobials such as macrolides by food-producing animals is independently correlated with AMR in humans (Mitchell et al., 1998; González et al., 2022; Kenyon, 2022). For example, an integrated analysis of the consumption of antimicrobials revealed positive association between macrolide use in animal farming and macrolide resistance in Campylobacter spp. in humans (European Centre for Disease Prevention and Control (ECDC), European Food Safety Authority (EFSA), European Medicines Agency (EMA), 2017). Another study has found that antibiotic consumption in animals was linked to resistance in some priority human pathogens (Allel et al., 2023). A related investigation found that reducing the use of macrolides in food-producing animals was associated with a decrease in bacterial macrolide resistance in both animals and humans (Tang et al., 2017). Another ecological study found a positive association between the intensity of macrolides used for food-producing animals and the prevalence of macrolide resistance in Streptococcus pneumoniae (Kenyon, 2022).

These findings provided the rationale for the central research question tested in this paper: can the amount of erythromycin allowed in food induce erythromycin resistance in S. pneumoniae? We use an in vivo model to test if the acceptable daily intake (ADI) of erythromycin, according to the European Medicines Agency (EMA) and World Health Organization (WHO)/Food and Agriculture Organization (FAO) is able to induce resistance to erythromycin. The ADI is defined by the FAO/WHO as “an estimate of the amount of a food additive in food or beverages expressed on a body weight (bw) basis that can be ingested daily over a lifetime without appreciable health risk to the consumer” (FAO/WHO, 2014). The ADI of a medicinal compound is based on studies evaluating thresholds for different types of toxicity (Murray et al., 2021; Subirats, Domingues & Topp, 2019). For the macrolides, EMA ADIs are determined based on microbiological toxicity (EMEA, 1998, 2000, 2007). These are established by evaluating the MICs for human bacterial commensal species and calculating estimated dose exposures in the human colon (EMA, 2019; FAO/WHO, 2021). EMA and WHO/FAO use the ADIs and other information such as dietary exposure to the relevant foodstuff to set maximum residue limits (MRLs). The WHO/FAO define the MRL as “the maximum concentration of residue resulting from the use of a veterinary drug (expressed in mg/kg or μg/kg on a fresh weight basis) that is recommended by the CAC to be legally permitted or recognized as acceptable in or on a food” (FAO/WHO, 2023). The MRL is set at a level that ensures that the residues in food do not exceed the ADI (FAO/WHO, 2014; EMA, 2019; FAO/WHO, 2023). In their assessments of ADIs and MRLs of antimicrobials in food products, the EMA and WHO/FAO guidelines do not include induction of resistance at sub-MIC concentrations (FAO/WHO, 2023; WHO, 2006).

The most recent EMA reports concluded that the ADI of erythromycin is 5 μg/kg/body weight (bw) (EMEA, 2000, 2002). The Joint FAO/WHO Expert Committee on Food Additives (JECFA) set a lower erythromycin ADI of 0.7 μg/kg/bw (WHO, 2006). We hypothesized that both these doses could induce erythromycin resistance in vivo. We tested this hypothesis in a Galleria mellonella model of chronic S. pneumoniae infection treated with ADI equivalent concentrations of erythromycin. Several studies have established that G. mellonella offers a useful model of human-microbial interactions (Andrea, Krogfelt & Jenssen, 2019; Cutuli et al., 2019).

Materials and Methods

Bacterial strain and live microbial inoculum preparation

Streptococcus pneumoniae strain (ATCC 49619) with a low erythromycin MIC (0.064 mg/L) was selected for the experiment. This strain was isolated from the sputum of a 75-year-old human male in Arizona. This strain is moderately penicillin-resistant and is used as the quality control strain in susceptibility testing (ATCC, Manassas, VA, USA). The selected strain of S. pneumoniae was cultured from the frozen stock onto Mueller Hinton Agar (MHA) + 5% horse blood (bioMérieux) at 37 °C overnight. Single colonies were selected from this culture and spread onto fresh agar plates, which were incubated at 37 °C with 5% (v/v) CO2 overnight. The S. pneumoniae was then inoculated into the haemocoel of the G. mellonella larva (10 μL of PBS containing 5.5 × 105 CFU of S. pneumoniae). This dose of S. pneumoniae was determined based on experiments that established a dose that enabled recovery of the bacteria up to 6 days post-inoculation without an excessive mortality rate of the G. mellonella (data not shown).

Injection of G. mellonella larvae with S. pneumoniae and erythromycin

Two batches of the last larval stage G. mellonella (Terramania, Arnhem, The Netherlands) were used for the experiments. These are referred to as batch one and batch two. Only macroscopically healthy, non-discolored larvae weighing 300 to 400 mg were selected (Table S1). The larvae were placed into individual Petri dishes in groups of 10 per dish. The larvae were kept in an incubator at 37 °C with a 5% (v/v) CO2 atmosphere for the duration of the experiments. Each control and experimental condition consisted of one and two groups of 10 larvae for the batch one and two experiments, respectively. Five conditions were tested: (1) PBS control, (2) 0.025 ng erythromycin (0.1ADI WHO), (3) 0.25 ng erythromycin (1ADI WHO), (4) 1.75 ng erythromycin (1ADI EMA) and (5) 17.5 ng erythromycin (10ADI EMA; Fig. 1).

Figure 1 Visual scheme illustrating injection of G. mellonella larvae with S. pneumoniae followed 10–20 min later by various doses of erythromycin (experimental group) or PBS (control group).

Visual scheme illustrating injection of G. mellonella larvae in the last pro-leg with bacterial suspension of S. pneumoniae (5.5 × 105 CFU) followed 10-20 minutes later by various doses of erythromycin (ERM; experimental groups) or PBS (control group). ADI, Acceptable daily intake; WHO, World Health Organization; EMA, European Medicines Association (Figure produced in BioRender.com).

These EMA doses were calculated based on EMA acceptable daily intake (ADI) of erythromycin (5 μg/kg/body weight) (EMEA, 2000, 2002) which translates into a dose of 1.75 ng for 350 mg G. mellonella—the average weight of the larvae used in our experiment (Table S1). The JECFA/WHO ADI of erythromycin is 0.7 μg/kg/bw (WHO, 2006). This translates into a dose of 0.25 ng for 350 mg G. mellonella. For all experiments, a control group was included that received the same protocol—bacterial inoculation followed by 10 μL/larva of PBS. The larvae were injected in the last pro-leg with 10 μL of bacterial suspension followed 10–20 min later by four various doses of erythromycin injected into a different pro-leg (Erythromycin lactobionate, Amdipharm, Basildon, UK) using 0.3 mL U-100 insulin syringes BD Micro-Fine (Fig. 1). One syringe and needle were used for 10 larvae in each Petri dish.The experiments were first conducted on batch one larvae and then the same protocol was repeated at a later time point with the batch two larvae.

Retrieval of S. pneumoniae from G. mellonella

A total of 24 h after the injection of the bacteria and at 24 hourly intervals thereafter, the mortality of each group of larvae were assessed and 1 larva from each group of 10 larvae was randomly selected for extraction of hemolymph. This was continued for the duration of the experiments—6 days.

The larvae were placed at −70 °C for 60 s until no movements were observable. Afterwards, they were placed on a Petri dish, and an incision was made between two segments closest to the tail of the larva. Hemolymph was then extracted by applying pressure to extract it directly into 1.8 mL centrifuge tubes containing 100 μL PBS. The hemolymph from each larva was vortexed and plated into plates with and without erythromycin. As a result one larva per group of 10 larvae was tested for the colonization by S. pneumoniae (no erythromycin plate) and the emergence of erythromycin resistance (erythromycin plate). Modified CNA agar +5% horse blood + 0.0032 g/L Crystal Violet with or without 3 × MIC (0.192 mg/L) of erythromycin were used for these experiments.

Plates were then incubated at 37 °C in a 5% (v/v) CO2 for 24 h, and the grey alpha-hemolytic colonies were counted manually (Fig. 2). All the grey colonies with greenish alpha-hemolysis zone growing on the erythromycin plates and a random selection of up to two single colonies from the non-antibiotic plates were selected for identification via Matrix-Assisted Laser Desorption/Ionization-Time-of-Flight mass spectrometry (MALDI-TOF MS). Following EUCAST guidelines, erythromycin resistance was defined as erythromycin MIC >0.25 mg/L (EUCAST, 2023).

Figure 2 Visual scheme illustrating hemolymph extraction of G. mellonella larvae at 24 hourly intervals after injection during 6 days of experiment and retrieval of S. pneumoniae.

Visual scheme of study methodology illustrating hemolymph extraction of 1–2 randomly selected G. mellonella larvae from each group of 10 larvae at 24 hourly intervals after injection during 6 days of experiment and retrieval of S. pneumoniae on selective agar plates with or without erythromycin. Colonies count was performed manually (Figure produced in BioRender.com; CC-BY-NC-ND).

The mortality of the larvae (including those used for haemolymph extraction) was assessed daily. Larvae were considered dead when there were no signs of movement in response to external prodding (Cools et al., 2019). When each experiment was completed, the surviving and dead G. mellonella were kept at −70 °C overnight to sedate and euthanize them. They were then autoclaved at 121 °C for 15 min and disposed of.

MALDI-TOF MS species identification and e-test

Each bacterial isolate was spread on a steel target plate and covered with 1 μL of α-cyano-4-hydroxycinnamic acid (CHCA) matrix solution. After drying the target plate was loaded and read. The spectra were acquired in linear mode in a mass range of 2–20 kDa and then compared to the library. Results were classified as reliable or unreliable according to recommended cut-off values of 1.7 and 2 for validated results for the genus and species levels, respectively (Laumen et al., 2022).

If S. pneumoniae colonies were suspected on an erythromycin plate, then four colonies were randomly selected and cultured on Mueller-Hinton agar (MHA) supplemented with 5% horse blood (Becton Dickinson) and incubated for 18–20 h at 37 °C in a 5% (v/v) CO2. Subsequently, the species identity of these colonies were confirmed using MALDI TOF-MS, and E-tests (AB bioMerieux, Solna, Sweden) were performed to determine erythromycin MICs. The E-tests were performed on MHA (Becton Dickinson, Franklin Lakes, NJ, USA) supplemented with 5% horse blood plates incubated for 18–20 h at 37 °C in a 5% (v/v) CO2, following EUCAST guidelines (EUCAST, 2023). E-tests strips were placed on a freshly inoculated plate using an inoculum of 0.5–1 McFarland of S. pneumoniae and read according to the manufacturer’s instructions. Erythromycin MICs were also performed on four randomly selected colonies from the baseline strain and four randomly selected colonies from the PBS control group from day 4 (the last day that S. pneumoniae grew on the erythromycin plates in the other experimental groups).

Data analysis

Statistical analyses and data visualization, such as graphs and boxplots, were performed using GraphPad Prism® version 9.5.1 with Mann-Whitney or ANOVA tests that were used to compare groups, depending on Gaussian distribution. The Kaplan-Meier statistical method was used for survival analysis. P-value < 0.05 was considered statistically significant. Visual schemes of materials and methods were created with BioRender (https://www.biorender.com/).

Whole genome sequencing

Whole genome sequencing (WGS) of the following samples was carried out—ATCC 49619, 1EMA ADI-2802b, 1EMA ADI-0303c, 1WHO ADI-1403b, 1WHO ADI-1403c, Ctrl-B2, Ctrl-B4. These samples were selected as follows. Two isolates from each of the two conditions with S. pneumoniae growth on the erythromycin plates (1EMA and 1ADI) were randomly chosen. In addition, the baseline strain and two randomly selected isolates from the control group on day four were sequenced. The whole genome sequencing was outsourced to Eurofins, where the samples were processed as follows. In brief, genomic DNA was extracted using the QIAGEN extraction Kit (DNeasy® Blood &Tissue Kit (50); Qiagen, Hilden, Germany) and suspended in nuclease-free water (Sigma-Aldrich, St. Louis, MO, USA). Paired-end 150-bp indexed reads using Nextera XT DNA library prep kit were generated using Illumina technology according to the manufacturer’s instructions (Eurofins, Konstanz, Germany).

After the initial quality control by FastQC (https://github.com/s-andrews/FastQC) and trimming using trimmomatic (Bolger, Lohse & Usadel, 2014), the processed Illumina reads were de novo assembled with Shovill v1.0.4 (https://github.com/tseemann/shovill), which uses SPAdes v3.14.0 (https://github.com/ablab/spades), with the following parameters: --trim --depth 150 --opts –isolate. Annotation of antimicrobial resistance genes was performed with Prokka, v1.14.6 (Seemann, 2014). PfaSTer, a machine learning-based method, was used to identify the serotypes from the assembled S. pneumoniae genomes (Lee et al., 2023).

Abricate v1.01 (https://github.com/tseemann/abricate) was used to search for virulence and antibiotic resistance (AMR) genes from the genome assemblies (.fna files) using the vfdb (Chen et al., 2015) and card (Jia et al., 2016) databases, respectively (updated 2021 March 27), the results are reported when they achieved >99% coverage and identity with no gaps. Additionally, the genomes with evidence of horizontal gene transfer (HGT), BLASTN was carried out against non-redundant (nr) database from NCBI. Furthermore, the quality-controlled reads were mapped to the ATCC 49619 reference genome. The different single nucleotide polymorphisms (SNPs) were determined with a minimum coverage of 10X and minimum frequency of 35% using the variant detection tool implemented in CLC genomics Workbench V22. The raw reads are deposited under the BioProject ID PRJNA1011801.

Results

Isolation, purification and identification of isolates

Following the inoculation of Streptococcus pneumoniae strain ATCC 49619 into two batches of Galleria mellonella larvae and subsequent exposure to varying concentrations of erythromycin, the clones were isolated on modified CNA agar plates supplemented with 5% horse blood and 0.0032 g/L Crystal Violet, with or without 3× MIC (0.192 mg/L) of erythromycin. The colonies that were grey alpha-hemolytic indicative of S. pneumoniae were selected for further purification. Colonies from erythromycin and non-antibiotic control plates were subcultured on MHA plates supplemented with 5% horse blood to ensure the purity and selective enrichment of S. pneumoniae clones for subsequent analyses. Following species confirmation using MALDI TOF-MS, E-tests were used to determine the erythromycin MICs.

Colonization and number of colonies tested

Batch one—colonization but no emergence of resistance

The G. mellonella larvae were successfully colonized for 4 days with the S. pneumoniae strain (Fig. 3). S. pneumoniae was recovered from the hemolymph on selective agar plates without erythromycin for 4 days after inoculation. No S. pneumoniae colonies were recovered from the selective plates with erythromycin.

Figure 3 Colonization of batch one G. mellonella larvae with S. pneumoniae on 1–6 days after injections of S. pneumoniae followed by various concentrations of erythromycin or PBS (control).

Batch one results. Colonization of G. mellonella larvae with presumed S. pneumoniae on days 1 to 6 after infection with S. pneumoniae followed by administration of various concentrations of erythromycin or PBS (control). ((A) EMA ADI doses) Colony count of S. pneumoniae growth done manually on non-antibiotic plates after injection of erythromycin ADI doses defined by EMA: 1.75 ng (1ADI EMA) and 17.5 ng (10ADI EMA) ((B) WHO ADI doses). Colony count of presumed S. pneumoniae growth done manually on non-antibiotic plates after injection of erythromycin ADI WHO doses: 0.025 ng (0.1ADI WHO) and 0.25 ng (1ADI WHO).

Batch two—colonization and emergence of resistance

The G. mellonella larvae were successfully colonized for 5 to 6 days with the S. pneumoniae strain (Fig. 4). S. pneumoniae species could be recovered from the hemolymph upon culturing on a selective agar plate without erythromycin for 5 to 6 days after inoculation. The species identity of 1–2 colonies per agar plate were confirmed with MALDI-TOF MS (Table S2).

Figure 4 Colonization of batch two G. mellonella larvae with S. pneumoniae on 1–6 days after injections of S. pneumoniae followed by various concentrations of erythromycin or PBS (control).

Batch two results. Colonization of G. mellonella larvae with presumed S. pneumoniae on days 1 to 6 after infection with S. pneumoniae followed by administration of various concentrations of erythromycin or PBS (control). ((A) EMA ADI doses) Colony count of S. pneumoniae growth done manually on non-antibiotic plates after injection of erythromycin ADI doses defined by EMA: 1.75 ng (1ADI EMA) and 17.5 ng (10ADI EMA) ((B) WHO ADI doses). Colony count of presumed S. pneumoniae growth done manually on non-antibiotic plates after injection of erythromycin ADI WHO doses: 0.025 ng (0.1ADI WHO) and 0.25 ng (1ADI WHO). The bars display the standard error of the mean.

Mortality

As expected, there was no significant difference in the mortality rates of the larvae between the antibiotic- treated and control groups (data not shown). The cumulative number of dead larvae per group of ten gradually increased from 1–2 on the first day after injection to 7–8 on the sixth day.

Emergence of AMR

The emergence of erythromycin resistance was assessed via manually counting colonies of S. pneumoniae on the selective agar plates with erythromycin. No S. pneumoniae colonies were seen on the control plates with erythromycin, 0.1ADI WHO erythromycin or10ADI EMA erythromycin plates (Fig. 5). Resistant colonies emerged on 1ADI EMA erythromycin plates at days 1 and 4 (Tables 1, 2; Fig. 5). Resistant colonies also emerged on 1ADI WHO plates at day 1 only (Tables 1, 2; Fig. 5). The median erythromycin MIC of the 1 ADI EMA colonies from days 1 and 4 was 0.36 mg/L (IQR 0.125–1 mg/L; Tables 1, 2). These MICs were 2 to 15 times higher than the baseline MIC—0.064 mg/L (P-value 0.0048; Fig. 6). The median erythromycin MIC of the 1ADI WHO colonies was 0.25 mg/L (0.190–0.38 mg/L; Tables 1, 2). These MICs were 3 to 6 times higher than the baseline MIC—0.064 mg/L (P-value 0.0286; Fig. 6). The erythromycin MICs increased slightly by day 4 in the PBS control group but this increase was not statistically significant (Table 2).

Figure 5 Batch two results. Emergence of colonies of S. pneumoniae with elevated erythromycin MICs on erythromycin plates, 1–4 days after injection of S. pneumoniae followed by administration of various concentrations of erythromycin or PBS (control).

((A) EMA doses). Manual count of S. pneumoniae colonies on erythromycin plates after injection of erythromycin ADI doses defined by EMA: 1.75 ng (1ADI EMA) and 17.5 ng (10ADI EMA) ((B) WHO doses). Manual colony count of S. pneumoniae on erythromycin plates after injection of erythromycin ADI doses defined by WHO: 0.025 ng (0.1ADI WHO) and 0.25 ng (1ADI WHO). The bars display standard error of the mean.

Table 1 S. pneumoniae ribosomal protein mutations detected.

S. pneumoniae isolates with elevated erythromycin MICs with available whole genomes and ribosomal protein mutations detected.

Dose	Strain ID	Groups	Erythromycin injected (ng)	MIC (mg/L)	Day of experiment	Ribosomal protein mutations (rplA)	
				
1EMA ADI	2802b	Test	1.75	0.25	1	Asn34Lys	
1EMA ADI	0303c	Test	1.75	0.190	4		
1WHO ADI	1403b	Test	0.25	0.25	1		
1WHO ADI	1403c	Test	0.25	0.38	1	Asn34Lys	
No dose	ATCC 49619	Reference		0.064	0		
No dose	Ctrl-B2	Control		0.125	4		
No dose	Ctrl-B4	Control		0.094	4		

Table 2 Erythromycin MICs (mg/L) of S. pneumoniae colonies.

Erythromycin MICs (mg/L) of randomly selected S. pneumoniae baseline colonies, colonies growing no-erythromycin plates following PBS injection (PBS control) and colonies growing on erythromycin plates following EMA Acceptable Daily Intake (1ADI EMA) and WHO ADI (1ADI WHO) doses of erythromycin.

	Day	Colony 1	Colony 2	Colony 3	Colony 4	P-value#	
Baseline	0	0.064	0.064	0.094	0.064	Ref	
1ADI EMA	1	0.19	0.25	1	–	**a	
1ADI EMA	4	0.19	0.38	0.125	–	a	
1ADI WHO	4	0.19	0.25	0.38	0.19	*	
PBS control	4	0.125	0.094	0.094		NS	
Note:

#Mann-Whitney test used to assess for statistically significant differences between erythromycin MICs of each experimental group and the baseline MICs (NS not significant; *P < 0.05, **P < 0.005). aThe MICs for the day 1 and day 4 isolates of the 1ADI EMA were combined for this comparison. Ref, Reference group.

Figure 6 Erythromycin MICs distribution for S. pneumoniae baseline colonies and resistant colonies (1ADI EMA and 1ADI WHO).

Erythromycin MIC distribution for S. pneumoniae baseline (ATCC 49619) colonies and resistant colonies (1ADI EMA (days 1 & 4) and 1ADI WHO (day 4)) in batch two. Mean values with SD. Mann-Whitney test used to assess for statistically significant differences (*P < 0.05, **P < 0.005).

Whole genome sequencing

WGS revealed mutations in relevant ribosomal proteins in two of the isolates exposed to erythromycin that were not detected in the control groups: 102C>G (Asn34Lys) in rplA gene encoding the 50S ribosomal protein L1 (Table 1). In addition, non-synonymous mutations were detected in six hypothetical proteins, as well as Val324Leu in xerS and Ser203Arg in Putative TrmH family tRNA/rRNA methyltransferase (Table 3). A number of synonymous mutations were also detected in lytA_2, recG, hrcA, arylsulfatase and two hypothetical proteins. The list of relevant mutations found in all the strains can be found in the Supplemental Materials (Table S4).

Table 3 Mutations detected in S. pneumoniae.

Mutations detected in S. pneumoniae exposed to low dose erythromycin but not in unexposed controls.

Gene product	CDS/Gene	Strain	
		2802b	1403b	1403c	0303c	Ctrl_B2	Ctrl_B4	ATCC 49619	
Experimental condition	–	1EMA ADI	1WHO ADI	1WHO ADI	1EMA ADI	PBS control	PBS control	Baseline	
Erythromycin MIC (mg/L)	–	0.25	0.25	0.38	0.190	0.125	0.094	0.064	
Hypothetical protein	OPMNIGBM_00355	–	–	–	c.888A>C	–	–	–	
Hypothetical protein	OPMNIGBM_00536	–	–	–	p.Arg77Trp	–	–	–	
Tyrosine recombinase	xerS	–	–	–	p.Val324Leu	–	–	–	
Hypothetical protein	OPMNIGBM_00800	–	–	–	p.Ser121Gly	–	–	–	
Putative TrmH family tRNA/rRNA methyltransferase	OPMNIGBM_00823	–	–	–	p.Ser203Arg	–	–	–	
Hypothetical protein	OPMNIGBM_00925	–	–	–	p.Trp47Leu	–	–	–	
Hypothetical protein	OPMNIGBM_01263	–	–	–	p.Leu8Ser	–	–	–	
Hypothetical protein	OPMNIGBM_00216	–	p.Glu159*	–	NA	–	–	–	
Hypothetical protein	OPMNIGBM_00913	–	p.His44Asn	–	NA	–	–	–	
Arylsulfatase	OPMNIGBM_00351	c.750T>C	–	c.750T>C	NA	–	–	–	
Heat-inducible transcription repressor	hrcA	c.954C>A	–	c.954C>A	NA	–	–	–	
50S ribosomal protein L1	rplA	p.Asn34Lys	–	p.Asn34Lys	NA	–	–	–	
Hyothetical protein	OPMNIGBM_00258	p.Cys102Arg	–	–	NA	–	–	–	
ATP dependant DNA helicase	recG	c.837A>G	–	–	NA	–	–	–	
Hypothetical protein	OPMNIGBM_00430	p.Asp27Asn	–	–	NA	–	–	–	
Autolysin	lytA_2	c.108C>T	–	–	NA	–	–	–	

Serotyping, virulence and antimicrobial resistance genes

All the S. pneumoniae isolates belonged to the 19F serotype. The following virulence genes were identified in all the isolates: cbpD, cbpG, cps4A, cps4B, cps4C, cps4D, hysA, lytA, lytB, lytC, nanB, pavA, pce,pfbA, ply, psaA (Table S3A). RlmA(II), patA, patB, pmrA AMR genes were identified in all the isolates (Table S3B). Additionally, vanRC gene was identified in one isolate-1EMA ADI-2802b (Table 2). Further BLASTN of the flanking region of the vanC cluster from 1 EMA ADI-2802b isolate showed 100% identity to Enterococcus innesii (accession no: AP025635.1) and Enterococcus casseliflavus (accession no: LR607377.1).

Discussion

The lowest single dose of erythromycin at which we could induce de novo resistance of S. pneumoniae (Serotype 19F) in a G. mellonella model was 0.7 μg/kg/bw. This dose is classified as a safe dose that can be consumed daily according to the WHO. It is 5.3-fold lower than the erythromycin MIC for this strain. According to the FAO/WHO, this dose of erythromycin can safely be ingested by humans on a daily basis. The EMA classifies doses seven-fold higher than this as safe for daily consumption. We did not assess if lower doses could induce resistance and thus cannot exclude the possibility that the in vivo S. pneumoniae MSCdenovo for erythromycin is lower than this value. Because the MSCselect is typically lower than the MSCdenovo (Gullberg et al., 2014, 2011), future studies are required to assess the MSCselect in vivo.

To the best of our knowledge, these data represent the first in vivo assessment of MSCs. As already noted, several ecological level studies have found a link between macrolide consumption in food animals and AMR in human-associated bacteria (González et al., 2022; Kenyon, 2022). Our findings could therefor help explain the high prevalence of macrolide resistance in S. pneumoniae and other bacteria in some East Asian countries, that report high macrolide consumption in food animals but moderate macrolide consumption in humans (Mitchell et al., 1998; Kenyon, 2022; Felmingham, Cantón & Jenkins, 2007).

The S. pneumoniae isolate used for this experiment had the following virulence genes: cbpD, cbpG, cps4A, cps4B, cps4C, cps4D, hysA, lytA, lytB, lytC, nanB, pavA, pce, pfbA, ply, psaA that encode products important for adherence, colonization, invasion, and survival (Huang et al., 2022; Holmes et al., 2001; Romero-Steiner et al., 2003; Marion et al., 2012; Croney et al., 2013; Mellroth et al., 2014; Rai et al., 2016; Wren et al., 2017; Subramanian, Henriques-Normark & Normark, 2019; Zhao et al., 2019). Both intrinsic and acquired mechanisms affect susceptibility to a large variety of antibiotics (El Moujaber et al., 2017). The following intrinsic AMR genes were present: rlmA(II), patA, patB and pmrA. RlmA(II) encodes a methyltransferase gene and pmrA, a MFS-type efflux pump that confers resistance to tylosin, mycinamicin and ciprofloxacin (Gill, Brenwald & Wise, 1999; Piddock et al., 2002; Lee et al., 2004). PatA and patB encode half-ABC transporters that have been shown to be involved in fluoroquinolone resistance (El Garch et al., 2010; Garvey et al., 2011).

The vanA resistance locus in Enterococcus, consists of a cluster of seven genes: vanS, vanR, vanH, vanA, vanX, vanY, and vanZ (Arthur & Courvalin, 1993; Arthur et al., 1993) that confers resistance to vancomycin and teicoplanin. The vanR along with vanS are involved in response regulation and the expression of proteins accountable for detecting the extracellular presence of antimicrobial drugs and intracellular signalling (Monteiro da Silva et al., 2020; Clewell et al., 2014; Hong, Hutchings & Buttner, 2008). Interestingly one isolate 1EMA ADI-2802b (Day 1) had the vanRC, a vanA gene found in the vanC cluster that could have been acquired from Enterococcus innesiii. Previously, Enterococcus innesiii sp. nov., was isolated from Galleria mellonella and found to encode atypical vancomycin resistance genes (Gooch et al., 2021). Enterococcal species are the dominant bacteria in G. mellonella microbiome (Allonsius et al., 2019). Our findings could therefore be explained by the S. pneumoniae acquiring this vanRC gene from one of the enterococcal species in the G. mellonella microbiome. To the best of our knowledge, this vanRC gene has never been detected in S. pneumoniae before.

Macrolide resistance in S. pneumoniae typically emerges via either active efflux or target modification. Active efflux is mediated via the acquisition of the mef(A), mef(E), or mef(I) efflux pumps (Ambrose, Nisbet & Stephens, 2005; Schroeder & Stephens, 2016). None of the genes coding these proteins were present in our strain of S. pneumoniae. Some of the resistant strains acquired mutations in rplA and rplD that code for the L1 and L4 ribosomal proteins. Target modification in the genes encoding riboproteins L4 and L22 has been shown to result in macrolide resistance in S. pneumoniae and other bacteria (Tait-Kamradt et al., 2000a, 2000b; Canu et al., 2002). Mutations in L1 have not, as yet, been found to be causally associated with macrolide resistance in S. pneumoniae. L1 serves as a ribosomal protein to bind rRNA and as a translational repressor binding its mRNA (Nevskaya, 2005). Studies in Mycoplasma bovis and Stenotrophomonas maltophilia have found that mutations in both L1 and L4 proteins were independently associated with elevated macrolide MICs (Calvopiña, Dulyayangkul & Avison, 2020; Waldner et al., 2022). For example, a genome wide association study in Mycoplasma bovis identified nucleotide variants in the above proteins as independently associated with macrolide MICs (Waldner et al., 2022). Two of the strains with erythromycin resistance had acquired a 102C>G mutation in rplA that coded for a Asn34Lys change in L1. We could not find any evidence that this mutation is associated with macrolide resistance. It is however possible that this mutation, possibly in conjunction with other two synonymous mutations (arylsulfatase-750T>C, and hrcA 954C>A) found in both isolates may play a role. Further experiments are required to test this hypothesis. Because macrolides act by binding to the 23S rRNA, and ribosomal proteins do not directly interact with macrolides, mutations in ribosomal proteins frequently cause resistance via inducing conformational changes in the 23S rRNA (Schlünzen et al., 2001; Tu et al., 2005; Wilson, 2013). It is therefore possible that some combination of these mutations is responsible for the observed increases in erythromycin MICs.

Two previous studies of sub-MIC exposure to ciprofloxacin and ceftriaxone in E. coli have found that low dose antimicrobials selected for resistant isolates, but that in only a minority of isolates could the elevated MICs be explained by known resistance mechanisms (Ching & Zaman, 2020; Ching & Zaman, 2021). Both studies found novel mutations that could explain the increased MICs. These findings suggest that low dose antimicrobials may select for AMR via different pathways to high dose exposure. These mutations may act as stepping-stones to the future emergence of higher levels of resistance. For example, studies have found that transient mutations in ribosomal proteins including in L4, L22 and L34 can act as stepping-stones to higher level macrolide and fluoroquinolone resistance (Gomez et al., 2017; Laumen et al., 2021). Another common form of target modification is acquisition of a methylase enzyme, erm(A) or erm(B) that methylate key residues of the 23S rRNA (Ambrose, Nisbet & Stephens, 2005; Fyfe et al., 2016). These methylases were not detected in any of our isolates.

There were a number of other limitations to our study. We only assessed the effect of a single dose of four concentrations of a single antimicrobial on a single strain of S. pneumoniae. The infection model used was a chronic hemolymph infection in G. mellonella. It would be more relevant to determine if these low doses could induce resistance after oral ingestion in different mammalian models or humans. Our experiment only considered a single dose of erythromycin, whereas this dose of the antimicrobial could be ingested daily by humans. Furthermore, we did not include low doses of other antimicrobials, biocides or antidepressants, all of which have been shown to act synergistically in inducing AMR at low concentrations (Seiler & Berendonk, 2012; Jin et al., 2018). We only assessed the MSCdenovo and not the MSCselect, which is typically the lower of the two (Gullberg et al., 2014; González et al., 2022). One of the studies that evaluated the effects of exposure to residual levels of erythromycin on human intestinal epithelium found that this erythromycin resulted in increased intestinal permeability (Hao et al., 2019). We did not assess these effects. The dose of erythromycin given was based on a larva of 350 mg. The larvae weighed between 300 and 400 mg, meaning that the 300 mg larvae received up to 14,3% higher erythromycin concentration than that prescribed (Table S1). No resistant colonies emerged in the S. pneumoniae exposed to the highest concentration of erythromycin (10 ADI EMA). If this is not related to the relatively low number of larvae evaluated per time point per condition, we are unable to explain this finding. Erythromycin resistance only emerged in a small number of experiments, and in none of the experiments in batch one. This may be related to the lower colonization rate and time of colonization by S. pneumoniae in batch one, as well as the lower number of larvae used in batch one. Between-batch variations in infection parameters such as colonization rates, related to factors such as different ages and diets of the larvae, have been noted in previous studies (Andrea, Krogfelt & Jenssen, 2019; Desbois & Coote, 2011). We did not perform transcriptomic or proteomic analyses. We did not conduct the complementation experiments necessary to assess if the novel mutations are causally associated with elevations in erythromycin MIC. It may, initially, be considered improbable that resistant colonies only emerged on days one and four in the 1ADI EMA group. A plausible reason why they were not seen on days two and three is, however, due to the fact that individual larvae were sacrificed on each day. Our results could thus be explained by the resistant mutants only emerging in the larvae that were sacrificed on days one and four.

Stanton et al. (2020) found experimental evidence that macrolide concentrations below the MSC could significantly increase the persistence of resistant strains of bacteria. They termed the concentration above which a significant increase in persistence is observed, the minimal increased persistence concentration (MIPC) (Stanton et al., 2020). We found resistant isolates up to 4 days after exposure to erythromycin. Future experiments in G. mellonella could include the persistence of resistant isolates to assess MIPCs. Another innovation in the field has been the development of the SELection End points in Communities of bacTeria (SELECT) assay which defines the lowest observed effect concentration as the antimicrobial concentration during exponential growth phase where the net growth of a bacterial community is significantly reduced (Murray et al., 2020). These concentrations have thus far been tested in environmental samples such as sewage samples (Murray et al., 2020). It would be useful to assess the SELECT assay in in vivo models such as the G. mellonella model.

Macrolides are frequently used for food producing animals (Van Boeckel et al., 2017; Center For Veterinary Medicine, 2022). In the United States, 9% of all macrolides consumed are used for this purpose (Center For Veterinary Medicine, 2022). In some countries, this usage is increasing. In the United States, for example, the consumption of macrolides for food animals increased by 21% between 2020 through 2021 (Center For Veterinary Medicine, 2022). Thus, if our results are replicated in mammalians, then it may be prudent to include the induction of AMR in the criteria used to define ADIs and MRLs. Finally, if our findings are validated in a mammalian gut model, then the G. mellonella model of chronic infections could be used as a high throughput tool to test safe ADIs for other bug-drug combinations.

Supplemental Information

Supplemental Information 1 Supplemental Tables.

Supplemental Information 2 Raw data.

Additional Information and Declarations

Competing Interests

Author Contributions

DNA Deposition

Data Availability

We have no competing interests.

Yuliia Baranchyk conceived and designed the experiments, performed the experiments, analyzed the data, prepared figures and/or tables, authored or reviewed drafts of the article, and approved the final draft.

Zina Gestels performed the experiments, authored or reviewed drafts of the article, and approved the final draft.

Dorien Van den Bossche analyzed the data, authored or reviewed drafts of the article, and approved the final draft.

Saïd Abdellati performed the experiments, authored or reviewed drafts of the article, and approved the final draft.

Basil Britto Xavier analyzed the data, authored or reviewed drafts of the article, and approved the final draft.

Sheeba Santhini Manoharan-Basil performed the experiments, analyzed the data, authored or reviewed drafts of the article, and approved the final draft.

Chris Kenyon conceived and designed the experiments, performed the experiments, analyzed the data, prepared figures and/or tables, authored or reviewed drafts of the article, and approved the final draft.

The following information was supplied regarding the deposition of DNA sequences:

BioProject ID PRJNA1011801.

The following information was supplied regarding data availability:

The raw sequence reads are available at NCBI: PRJNA1011801.

https://www.ncbi.nlm.nih.gov/bioproject/?term=PRJNA1011801.

The raw data is available in the Supplemental Files.

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
