# Peer review of "Effect of erythromycin residuals in food on the development of resistance in Streptococcus pneumoniae: an in vivo study in Galleria mellonella"

_PeerJ, doi:10.7717/peerj.17463_

## Round 0.1 · original submission · Major Revisions

Dear Dr. Baranchyk and colleagues:

Thanks for submitting your manuscript to PeerJ. I have now received three independent reviews of your work, and as you will see, the reviewers raised some concerns about the research. Despite this, these reviewers are optimistic about your work and the potential impact it will have on research studying AMR in Streptococcus pneumoniae. Thus, I encourage you to revise your manuscript, accordingly, considering all the concerns raised by both reviewers.

Please revise your manuscript for clarity and limit jargon/reduce verbiage but be clear about the focus of the study and the target audience. Focus especially on clarity, and address the sections considered incomplete and/or unclear by the reviewers. It appears that certain key references are missing. The Methods should be clear, concise and repeatable. Please ensure this. Also, elaborate on the discussion of your findings, placing them within a broad and inclusive body of work by the field. Please supply any code or scripts in the supplemental material.

Please ensure all experiments are repeated, providing the number of replicates (both within individual experiments and separate experiments).

I look forward to seeing your revision, and thanks again for submitting your work to PeerJ.

Good luck with your revision,

-joe

·

Basic reporting

This paper is important, novel, and impactful. The authors did a great job with the study design, introduction, and discussion points. However, considerable improvements need to be made before publication, especially on the scientific writing (see comments below).

Experimental design

Experimental design could be better explained in the diagram they provide, better explaining the dosing, euthanasia, mortality, etc (see detailed comments below).

Validity of the findings

Finding are valid, but poorly explained. More details on the group differences in their figures are needed. It appears that they were able to isolate quite a bit of antimicrobial resistant bacteria after treatment, but they report MICs for just 2 (if I interpreted correctly) and performed WGS for just a few colonies as well. How were they selected among all the resistant-isolates you had?

Additional comments

Specific comments are below:


Line 30: infection should not be italicized.
Line 33: rephrase to say something like: erythromycin ADI doses (dose A, dose B) according to EMA and WHO, respectively.
Line 65: replace “-“ by a comma.
Line 114: It would be good to know more about this strain – if it has been shown to cause clinical disease, for example.
Line 119: Maybe use 5.5.x 102
Line 122: Are there papers that have used this same infection model that you could use to support your methods?
Line 123: make the subtitles more specific – injection with what
Line 125: 300-400mg of weight?

Line 129: Be more specific with the doses and volumes. It is not clear.
Paragraph beginning in line 134: use cleared language for your dosing. I can see what you mean, but it is definitely not clear. Maybe you could add in your figure1 diagram all the different doses for each group.

Figures 2 and 3 you need to specify what are panels A and B
Figures: it would be valuable to have the comparison of MICs between control, 1ADI, and 10ADI.
Line 142: Does this mean that the larvae were euthanized by placing in the freezer? At first it sounds like there was mortality due to experimental infection – please clarify.
Line 161: This is not scientific writing.
Line 172: I am not sure this is the best way to describe results from a E-test. E-tests are pretty specific to identify the exact MIC.
Line 180: better explanation of why and how each of these isolates were chosen to be sequenced is needed.
Line 191: Annotation of antimicrobial resistance genes was performed with Prokka
Line 192: use a comma after method.

Line 195: when instead of where
Line 197: what is nr database.
Line 206: identified?
Line 208: was mortality expected with your infection model? This needs to be explained.
Line 280: italicize genes
Line 288 -290: This phrase could just be incorporated with the above.
Line 296: capital 23S
Line 309: italicize genes.
Line 326-327: please rephrase.

·

Basic reporting

The manuscript is concisely and clearly written. Except of a few typos and some small phrasing issues, I did not see any problems with the language used.
In the introduction, the authors explain the importance of studying the effects of antibiotics present in food. The background includes low minimal selective concentrations (MSCs) that have been reported in the literature and the association of antibiotics consumption in food-producing animals to AMR in humans. Structure of the article and references are fine.

Experimental design

The manuscript presents original primary research within the scope of the journal. The hypothesis is clearly stated and the importance of the research has been described.

Galleria mellonella is a well characterized invertebrate model for the study of bacterial and fungal infections. It is suited for this particular study, since the scientific aim was not focusing on the specific host immune response but on the low doses exposure of bacteria in a complex environment.
It is not clear to me, whether and how the colonies grown on antibiotic plates were isolated, before the erythromycin MICs were evaluated. The number of passages, before MIC determination gives information about the stability of the mutation. Because I have some doubts about the validity of the findings, I will go into detail in the next section.

Validity of the findings

• 3. Validity of the findings
G. mellonella survival rates were typical for the infection dose of S. pneumoniae used in this study. S. pneumoniae could be re-isolated from infected larvae and erythromycin resistant bacteria were only detected in hemolymph extracted from treated larvae. The range of erythromycin concentration that seemed to be able to induce de novo resistance in S. pneumoniae under these conditions was 0.25-1.75 ng/larva, corresponding to 1ADI WHO and 1ADI EMA, respectively. Administration of lower or higher doses of erythromycin (0.025 ng/larva or 17.5 µg/larva) did not induce development of resistance. In resistant clones, several SNPs were identified that might be correlated to resistance. Overall, this is a result with a certain plausibility. However, I have a lot of questions about some details:

• Lines 224-226: < Resistant colonies emerged on 1ADI EMA erythromycin plates at days 1 and 4 (Table 1; Fig. 4). Resistant colonies also emerged on 1ADI WHO plates at day 1 only (Table 1; Fig. 4).>
In the case of 1 ADI WHO, resistant colonies emerge exclusively on day one, which could be explained by the protocol: only one dose of erythromycin, dilution effects, lower fitness of resistant clones in comparison to the WT bacteria etc. But how is it possible that resistant colonies (1ADI EMA) emerge on days one and four? This is hard to believe.

• Lines 226-228: < The median erythromycin MIC of the 1 ADI EMA colonies was 0.36 mg/L (IQR 0.125 - 1 mg/L; Table 1). These MICs were 2 to 15 times higher than the baseline MIC ñ 0.064 mg/L (P-value 0.0048; Fig. 5).>
A systematic information about how many clones were picked and how they were isolated before MIC evaluation is missing. A summary table showing the number of isolates with the corresponding MIC information and comparison to the reference strain and the two control strains needs to be provided. Even though the baseline MIC is low with 0.064 mg/L, the two controls are already elevated and no information is given about how many colonies of the control samples have been investigated.

In Table two, no MICs are indicated for the strains shown. Also a I am missing a nomenclature of the isolated strains.

The number of S. pneumoniae colonies on agar plates without antibiotics was about 170 (day one 1ADI EMA) and 180 (day four 1ADI EMA) (Figure 3). On agar plates with erythromycin (Figure 4), approximately 70 and 60 colonies were identified, respectively. This seems to be a very high number of resistant colonies. The authors do not relate these numbers and do not give any explanation.

Additional comments

In the results part of the manuscript, a detailed description of the overall protocol of isolating the resistant clones, their nomenclature, the purification steps, and the number of colonies tested has to be added. Also it is not clear to me, how many times the experiment has been repeated and whether the occurrence of resistant clones war repeatedly observed at days 1 and 4. Furthermore, a table with the respective MICs in comparison to reference and control colonies has to be added. The way, the data are presented in the current version of the manuscript, I am not able to judge whether the data are meaningful. In particular, there is no interpretation of the sporadic emergence of resistant clones, which did not seem to follow a logical sequence.

Even though the subject of this study is extremely important, and the general setup of the experiment seems to work, the analyses, presentation, and interpretation of the data do not support the conclusion of the study. At this point, I do not believe that any discussion about the mutations detected in the clones is useful.

Reviewer 3 ·

Basic reporting

- Line 43: grammar singular/plural
- Line 68-70: It is unclear what microorganisms you are referring to concerning the predicted MSC value and for which the MSC of erythromycin has and has not been evaluated.
- Line 128 mentions 20 larvae per group; line 144 mentions 10 larvae per group. It is unclear if this experiment has been performed on 10 larvae twice, or on 20 larvae once. Given the variation that exists within G. mellonella populations, in part due to the lack of standardization and vendors for scientific use, a repeat is important. Moreover, it is known these larvae tend to be treated with antibiotics prior to being sold, which could affect their microbiota. Exposing a second batch of larvae to the same experiment might have given very different results, if this was not done.
- Line 241 – 243: The numbers between parentheses are confusing. I’d add them to the materials and methods part and leave them out here. Also, the comparison between the MIC and the concentration of the injected volume is too straight forward and excludes the volume of the larval hemolymf. The average volume of the collected hemolymf could provide better insight in the actual concentration of erythromycin in the larvae.
- Figure 3 implies all recovered bacteria are S. pneumoniae. However, without antibiotics there will be a multitude of, potentially hemolytic, bacteria recovered from the hemolymf, as shown by your MALDI data. I find the text with this figure a bit too ambiguous.
- The extent to which raw data have been shared is impressive. It’s not often there is full transparency on larval weights, colony counts… as this is often seen as redundant. However, this gives a good insight in the experimental setup and the way of collecting and processing data.

Experimental design

- Why has the ADI been chosen to test the hypothesis, rather than a range of MRL values?
- From an experimental stance, I recommend injecting the bacterial suspension and the antibiotic solution in a different proleg, to avoid the possibility of creating an injury twice.
- Have the authors checked if their freezing step didn’t have a negative effect on bacterial survival? For example, have the authors injected a larvae with a known inoculum, directly followed by the retrieval procedure to ensure bacterial survival in these harsh conditions? The emergence of resistance only on day 1 and on day 4, but always in both larvae of the same group is of concern. As antibiotics were only administered on the first day, there is no reason why resistant bacteria that emerged already at day 1 – in both larvae – should not still be detectable on later timepoints. The re-emergence on day 4 is a possible indication of a wrong recovery in the earlier days, rather than a new rise in resistance. Again, an independent repeat of the experiment could provide insight in this observation.
- Is the probability of survival based on a separate survival experiment, or does this also include the larvae that were taken out for MIC analysis? If this is based on a separate analysis, this is an interesting find especially as your colony count does not change (a lot) during the course of the experiment. Have more research been done whether or not the recovered bacterial population changes over time? The MALDI results show no indication of this, but the number of colonies that were tested is rather limited. Lastly, the probability of survival on the sixth day is mentioned as 7-8 per 10 (Line 211), but SFig1 shows a 100% death rate on day 6.
- It is unclear how the strains were selected for WGS, and from which days they originate. Also, the MALDI identification data for the antibiotic plates are not included in the supplementary material.

Validity of the findings

- The authors were critical towards their own study, showing the need for studies like theirs while also acknowledging current limitations and suggesting follow-up research. It will be interesting to see how this research will be continued, as per the authors’ suggestions with more complex models and a more extensive test panel.

Additional comments

- This is a well thought-through research question. Especially in light in rising AMR and the lack of near-future alternatives, attention should be brought to the use of antibiotics in food or environmental applications. However, it is unclear if an independent repeat has been performed and if not, this is important given the nature of the model that was used.
- Both in the introduction and discussion more attention could be brought to the studies in wastewater, where for example (Stanton, 2020) found MSCs for macrolides were higher than those found in the environment – as opposed to the studies referenced by the authors.
- If not performed, an independent repeat of the vivo part (excluding WGS) is imperative to ensure no wrong conclusions of re-emergence or overall emergence of resistance are made. If an independent repeat was performed, this should be made more clear.

---

## Round 0.2 · Minor Revisions

Dear Dr. Baranchyk and colleagues:

Thanks for revising your manuscript. The reviewers are mostly satisfied with your revision (as am I). Great! However, there are edits to make and a few more issues to address. Please tend to these matters these ASAP so we may move towards acceptance of your work.

Best,

-joe

·

Basic reporting

The authors addressed well the concerns from me and other reviewers. A few more corrections are needed before accepting this paper.
Abstract conclusion – maybe add the importance of the mutation you found

Experimental design

Line 52 - Salmonella spp. the “spp.” should not be italicized
Line 83: replace hypothesis by research question. The question presented is not a true testable hypothesis.
Line 84: You don’t need to say “more specifically”.
Line 151: please, find a replacement for the word “kill” for larvae.

Validity of the findings

Line 163: identify should be identified
Line 235: replace “everyday life” with something more specific.

Additional comments

none.

·

Basic reporting

Basic reporting is fine

Experimental design

Experimental design has been explained now satisfactorily

Validity of the findings

Due to the changes in the manuscript, the validity of the findings can be assessed. The limititations of the study are stated and the data support the conclusions.

Reviewer 3 ·

Basic reporting

- Please re-check all references to figures/tables. Sometimes “Fig. X” is used, sometimes “Figure X” is used. Referencing to Tables in WGS- and serotyping-part does not seem correct.
- SFig1: The control group mortality differs between A and B – why?
- Stable2b: Please re-check data on 1ADI EMA and 1ADI WHO as day 4 seems incorrect.
- Figure 3: What is on the x-axis beyond day 6? The lay-out in between graphs is inconsistent.
- Figure 3-4: These figures don’t show the individual datapoints as does figure 5. Please change to also show these.
- Figure 5: Why is there no day 6 on this graph?
- In general, there are several issues with spacings throughout the manuscript (double spacings, no spacing after a point…).
- L182: “If S. pneumoniae colonies were observed on an erythromycin plate…” I’d rephrase this to “If S. pneumoniae colonies were suspected on an erythromycin plate…” as your MALDI-TOF shows not all growth that was presumed to be S. pneumoniae actually is S. pneumoniae (STable 2b: 1ADI WHO day 1).
- Figure 1: Thank you for clarifying the injection in the pro-legs. To be consistent with the text, I’d also change the figure to show the different injection sites.

Experimental design

- Overall, the experimental part about selection of colonies for identification with MALDI-TOF remains very unclear to me. For example, L182-183 states only 4 colonies were selected for MALDI-TOF, yet STable2 1ADI WHO has a total of 5 identifications. Also, the “retrieval of S. pneumoniae from G. mellonella” section (L164-167) reads “All the grey colonies with greenish alpha-hemolysis zone growing on the erythromycin plates and a random selection of up to two single colonies from the non-antibiotic plates were selected for identification”, which implies far more than 4 colonies were identified. Later (L248-249), it is stated that “the species identity of 1-2 colonies per agar plate were confirmed with MALDI-TOF”.
- I’m unclear how many larvae were euthanized every 24 hours to assess their bacterial burden. L151-152 state 1 larvae per group of 10 is euthanized every 24 hours. Does that mean for the second batch, which has 20 larvae per treatment group, 2 larvae per 24 hours were euthanized every time? If so, I don’t see this in SFig 1 (e.g. the control only has 1 death in the first 24 hours in SFig 1A). If not, please write this more clear.

Validity of the findings

No additional comments.

Additional comments

Thank you for the clarifications. While there remain some unclarities in the experimental section, the overall conclusions of the study and its limitations are well-rounded and self-critical. When these are resolved, this study will be a great addition to the rather limited number of research papers on G. mellonella as an infection model – which will help in establishing this model as a valid alternative to conventional animal models.

---

## Round 0.3 · accepted · Accept

Dear Dr. Baranchyk and colleagues:

Thanks for revising your manuscript based on the concerns raised by the reviewers. I now believe that your manuscript is suitable for publication. Congratulations! I look forward to seeing this work in print, and I anticipate it being an important resource for groups studying AMR in Streptococcus pneumoniae. Thanks again for choosing PeerJ to publish such important work.

Best,

-joe